# The Validation of the Danish Version of the Santa Barbara Sense of Direction Scale (SBSoDDK) and a Comparison of Performance on the SBSoD Across Samples with Different Nationalities

**DOI:** 10.3390/bs15030334

**Published:** 2025-03-09

**Authors:** Maria Beck Gaarde, Christian Gerlach

**Affiliations:** Department of Psychology, University of Southern Denmark, Campusvej 55, DK-5230 Odense, Denmark; cgerlach@health.sdu.dk

**Keywords:** cross-cultural comparison, mental maps, scale validation, self-report scale, sense of direction

## Abstract

The Santa Barbara Sense of Direction scale (SBSoD) is a self-report scale that assesses the ability to orient oneself in the environment. With the current study, we validated the Danish version of the SBSoD (SBSoDDK) and compared performance on the SBSoD across samples with different nationalities. We collected data for a Danish sample consisting of 119 Danish university students and received data from studies validating other versions of the SBSoD using American, German, Japanese, Chinese, Turkish, and British samples. The internal consistency and convergent and discriminant validity of the SBSoDDK, which exhibited a two-factor structure, were affirmed, and differences in performance on the SBSoD across samples with different nationalities were identified and may be linked to cultural variations in the sense of direction. The current study has certain weaknesses, including using nationality as a proxy for culture and variations in the gender compositions of the samples. Future studies should focus on addressing the current study’s weaknesses and revising and further validating the SBSoD.

## 1. Introduction

The sense of direction is the ability to orient oneself in the environment by aligning one’s position and direction with the position and direction of elements in the environment, such as landmarks and cardinal directions ([16]). The Santa Barbara Sense of Direction scale (SBSoD) is a self-report scale that assesses the sense of direction. The American English version of the SBSoD (SBSoDUS) is the original one and has been translated into several languages. Different studies ([22]; [30]; [6]) have validated the SBSoDUS as well as the German, Japanese, Chinese, Turkish, and British English versions of the SBSoD (SBSoDDE, SBSoDJP, SBSoDCN, SBSoDTR, and SBSoDGB). The primary aim of the current study is to validate the Danish version of the SBSoD (SBSoDDK) with a focus on reliability, validity, and factor structure. Convergent validity is evaluated by comparing performance on the SBSoDDK and the Four Mountain Test (4MT), which assesses topographical perception and memory ([15]; [19]), and discriminant validity is evaluated by comparing performance on the SBSoDDK and the Cambridge Face Memory Test (CFMT), which assesses face recognition ([8]). The secondary aim of the current study is to compare performance on the SBSoD across samples with different nationalities, which we, in the current study, use as a proxy for culture. Because self-report scales require minimal resources, and individuals can generally judge their sense of direction well ([2]), the SBSoD is suitable for cross-cultural comparisons of the sense of direction.

In relation to the sense of direction, a distinction is made between route and survey strategies. Route strategies rely on knowledge of landmarks and their connections and can be exemplified as “First, turn right at the church, then, turn left at the big apple tree, and, finally, turn right at the yellow house”. In comparison, survey strategies rely on mental maps, which resemble conventional maps but with less detail and precision. Survey strategies are generally considered superior to route strategies ([25]). The utilization of route and survey strategies appears to vary across cultures. For instance, a study by [33] ([33]) showed that words from the relative system (e.g., left), which change meaning depending on one’s position and direction, are included more frequently in route directions provided on websites for hotels in Great Britain compared to the United States. Moreover, a study by [22] ([22]) found that words from the absolute system (e.g., south), which do not change meaning depending on one’s position and direction, occur two to four times more frequently in American English and Chinese corpora than in German ones. Words from the relative system are associated with route strategies, whereas words from the absolute system are associated with survey strategies ([21]). Therefore, variations in the utilization of route and survey strategies across cultures may be related to the use of words from the relative and absolute systems. Furthermore, in some cultures, including China and so-called New World countries, such as the United States, the environment tends to be structured. Conversely, in other cultures, including most European and Asian countries, the environment tends to be unstructured ([1]; [4]). This may have implications for the sense of direction. For example, [10] ([10]) proposed that mental maps are formed more easily in a structured environment than in an unstructured one, and a study by [24] ([24]) demonstrated that American children typically include cardinal directions in drawings of the environment. Since mental maps are partly based on cardinal directions ([25]), this is consistent with [10]’s ([10]) proposal. Thus, variations in the utilization of route and survey strategies across cultures may also be related to the structure of the environment. Figure 1 illustrates an example of a structured and unstructured environment, respectively.

## 2. Materials and Methods

We published a preregistration ([12]) on the Open Science Framework prior to data analysis.

### 2.1. Sample

For the current study, a sample consisting of Danish university students was used. This was advantageous since the samples that were used for the studies ([22]; [30]; [6]) validating other versions of the SBSoD also consisted of university students (with different nationalities). Before data collection, it was clearly stated that participation was voluntary; consent was implied through participation; and withdrawal was allowed at any time. One individual withdrew, and 17 individuals were excluded due to incomplete data collection. As a result, the size and gender composition of the Danish sample were 119 and 70% females and 30% males, respectively. Moreover, the Danish sample consisted of 89% right-handed, 4% left-handed, and 7% ambidextrous individuals. Additional demographic data, such as age and ethnicity, were not collected to ensure anonymity. However, based on prior experience with a similar sampling approach, the expected age range of the Danish sample was between 20 and 50, with the majority (>90%) falling within the 20–25-year age group ([11]).

### 2.2. Data Collection

Within five days, we collected data for the following variables for approximately one-fourth of the Danish sample at a time in a consistent order:SBSoDDK: The scores for the 15 SBSoDDK items;4MT: The number of correct trials for the 4MT (range: 0–48);CFMT: The number of correct trials for the CFMT (range: 0–72).

The data were collected as part of a course offered by the University of Southern Denmark, which also approved the current study. According to Danish law, no additional approval was necessary.

#### 2.2.1. SBSoDDK

For the current study, the SBSoDUS was translated into Danish by three researchers with Danish as their native language. Initially, the researchers independently translated the SBSoDUS into Danish, intending to use easily comprehensible and accurate words. Subsequently, they compared and synthesized their translations into the SBSoDDK (see Section A.1).

The SBSoD items address various capacities, attitudes, and experiences related to the sense of direction. Three of the SBSoD items address the capacities for and attitude toward giving and understanding directions (numbers 1, 8, and 11); two of the SBSoD items address the capacity for and attitude toward reading maps (numbers 7 and 9); and two of the SBSoD items address the capacity for and experience with remembering (new) routes (numbers 10 and 14). The remaining SBSoD items address the capacities for finding one’s things (number 2), judging distances (number 3), orienting oneself in the environment (number 4), navigating in a new environment (number 6), and forming mental maps (number 15); the attitude toward knowing one’s position (number 12); and the experiences with using cardinal directions (number 5) and navigational planning (number 13). Some of the SBSoD items are more representative of route strategies than of survey strategies and vice versa. Item numbers 2, 6, 10, 12, and 14 are more representative of route strategies than of survey strategies, while item numbers 5, 7, 9, and 13 are more representative of survey strategies than of route strategies. Item numbers 3 and 4 can be representative of both route and survey strategies. As previously highlighted by [6] ([6]), directions can be understood as both route and cardinal directions, and mental maps can also be understood as knowledge of landmarks and their connections. Therefore, item numbers 1, 8, 11, and 15 can also be representative of both route and survey strategies.

The Danish sample (and the remaining samples) was asked to score the SBSoD items on a Likert scale ranging from 1 (strongly agree) to 7 (strongly disagree). The scores for the SBSoD items where a high score reflected a poor sense of direction (numbers 1, 3, 4, 5, 7, 9, and 14) were reversed. This allowed the scores for the SBSoD items to be meaningfully aggregated into a total SBSoD score. Thus, a high score for the SBSoD items and a high total SBSoD score reflected a good sense of direction.

#### 2.2.2. 4MT

The 4MT is a computer-based test that assesses topographical perception and memory. In the 4MT, the stimuli consist of images depicting semicircular landscapes of four mountains from varying perspectives. Together, the topographical perception and memory subtests of the 4MT (4MTper and 4MTmem) had 51 trials, including three practice trials. Five images were presented in each trial: one study image, one target image, and three distractor images. In the 4MTper, the study, target, and distractor images were presented concurrently until a key was pressed or 60 s had elapsed. In the 4MTmem, the study image was presented in isolation followed by a two-second delay of a black screen with a fixation cross. Then, the target and distractor images were presented until a key was pressed or 60 s had elapsed. The Danish sample was asked to identify the target image among the target and distractor images by pressing keys 1, 2, 3, or 4 on a keyboard ([15]; [19]).

#### 2.2.3. CFMT

The CFMT is a computer-based test that assesses face recognition. In the CFMT, the stimuli consist of images depicting faces. The CFMT had 75 trials, including three practice trials. Four images were presented in each trial: one study image, one target image, and two distractor images. The study image was presented in isolation before the target and distractor images were presented. In 18 of the trials, the target and study images were identical in pose and lighting; in 30 of the trials, the target images differed from the study images in pose and/or lighting; and in 24 of the trials, the target images differed from the study images in pose and/or lighting and Gaussian noise was added to the target images. The Danish sample was asked to identify the target image among the target and distractor images by pressing keys 1, 2, or 3 on a keyboard ([8]).

### 2.3. Hypotheses

In the preregistration ([12]), we listed the following hypotheses (or expectations) to be tested:

**Hypothesis** **1****(**SBSoDDK—internal consistency**).** *Comparable to other versions of the SBSoD, Cronbach’s alpha for the SBSoDDK will reach at least an acceptable level, which is 0.70 according to [3] ([3]).*

**Hypothesis** **2****(**SBSoDDK—item–total correlations**).** *The correlations between the scores for each SBSoDDK item and the total SBSoDDK scores will be positive and significant.*

**Hypothesis** **3****(**4MT and CFMT—internal consistency**).** *Cronbach’s alphas for the 4MTper, 4MTmem, and CFMT will reach at least acceptable levels.*

**Hypothesis** **4****(**SBSoDDK—convergent validity**).** *The correlation between the total SBSoDDK scores and the number of correct trials for the 4MTper and 4MTmem will be positive, significant, and weak to moderate.*

**Hypothesis** **5****(**SBSoDDK—discriminant validity**).** *The correlation between the total SBSoDDK scores and the number of correct trials for the CFMT will not be significant.*

**Hypothesis** **6****(**SBSoDDK, 4MT, and CFMT—comparison of correlations**).** *The correlations between the total SBSoDDK scores and the number of correct trials for the 4MTper and 4MTmem will be significantly stronger than the correlation between the total SBSoDDK scores and the number of correct trials for the CFMT.*

**Hypothesis** **7****(**SBSoDDK—factor structure; SBSoD versions—comparison of correlations**).** *(1) Comparable to other versions of the SBSoD, the SBSoDDK will exhibit a one- or two-factor structure; (2) the correlations between the factor loadings for the different SBSoD versions will be positive, significant, and moderate to strong; (3) the correlations between the mean scores for the items of the different SBSoD versions will be positive, significant, and moderate to strong; and (4) the mean total scores for the different SBSoD versions will be significantly lower for females than for males.*

### 2.4. Data Analysis

For data analysis, we used IBM SPSS Statistics version 28.0.0.0. For hypotheses 1 and 3, we calculated Cronbach’s alphas for the SBSoDDK, 4MTper, 4MTmem, and CFMT. For hypothesis 2, we calculated Pearson’s r correlations between the scores for each SBSoDDK item and the total SBSoDDK scores. For hypotheses 4 and 5, we calculated Pearson’s r correlations, both adjusted and non-adjusted ones, between the total SBSoDDK scores and the number of correct trials for the 4MTper, 4MTmem, and CFMT. The adjusted Pearson’s r correlations were calculated based on [28]’s ([28]) formula. For hypothesis 6, we compared the Pearson’s r correlations between the total SBSoDDK scores and the number of correct trials for the 4MTper and 4MTmem and the Pearson’s r correlation between the total SBSoDDK scores and the number of correct trials for the CFMT via the Fisher r-to-z transformation. For hypothesis 7, we (1) conducted an explorative factor analysis (EFA) for the SBSoDDK with principal axis factoring as the extraction method, (2) calculated Pearson’s r correlations between the factor loadings for the different SBSoD versions, (3) calculated Pearson’s r correlations between the mean scores for the items of the different SBSoD versions, and (4) performed independent t-tests for the mean total scores for the different SBSoD versions for females and males. All of this was possible because we received data from studies ([22]; [30]) validating the SBSoDUS, SBSoDDE, SBSoDJP, SBSoDCN, and SBSoDTR. Shapiro–Wilk normality tests indicated that the data deviated significantly from a normal distribution. However, we still performed the abovementioned data analysis as the sizes of the samples exceeded 30 ([14]). Since Bartlett’s test of sphericity exhibited a significant homogeneity of variance, and the Kaiser–Meyer–Olkin measure of sampling adequacy was above 0.60, conducting an EFA for the SBSoDDK was warranted. To ensure optimal comparability, we conducted the EFA for the SBSoDDK in the same manner as [6] ([6]) conducted an EFA for the SBSoDGB.

## 3. Results

The sizes and gender compositions of the American, German, Japanese, Chinese, Turkish, and British samples are shown in Table 1.

Cronbach’s alpha for the SBSoDDK was 0.89. As shown in Table 2, this is comparable to the SBSoDUS, SBSoDDE, SBSoDJP, SBSoDCN, SBSoDTR, and SBSoDGB. Furthermore, Cronbach’s alphas for the 4MTper, 4MTmem, and CFMT were 0.49, 0.65, and 0.88, respectively.

As Table 3 shows, the correlations between the scores for each SBSoDDK item and the total SBSoDDK scores were positive, significant, and moderate to strong.

The correlations between the total SBSoDDK scores and the number of correct trials for the 4MTper and 4MTmem were positive, significant, and weak to moderate; r_adj_ = 0.29, r = 0.19, *p* = 0.02 (one-tailed) for the 4MTper, and r_adj_ = 0.37, r = 0.28, *p* < 0.001 (one-tailed) for the 4MTmem. The correlation between the total SBSoDDK scores and the number of correct trials for the CMFT was not significant; r_adj_ = −0.14, r = −0.12, *p* = 0.21 (two-tailed). Moreover, the correlations between the total SBSoDDK scores and the number of correct trials for the 4MTper and 4MTmem were significantly stronger than the correlation between the total SBSoDDK scores and the number of correct trials for the CFMT; z_adj_ = 3.60, z = 2.50, *p* = 0.006 (one-tailed) for the 4MTper and CFMT, and z_adj_ = 4.26, z = 3.24, *p* < 0.001 (one-tailed) for the 4MTmem and CFMT.

The EFA for the SBSoDDK revealed eigenvalues greater than 1.0 for three factors: 6.2 for factor 1, 1.5 for factor 2, and 1.1 for factor 3. However, in the scree plot, only factors 1 and 2 were beyond the “breakpoint”. Hence, the SBSoDDK exhibited a two-factor structure, which explained 51.6% of the variance (41.4% for factor 1 and 10.2% for factor 2). This is comparable to the SBSoDGB since it, as shown in Table 2, also exhibited a two-factor structure, which explained 47.0% of the variance (36.6% for factor 1 and 10.4% for factor 2). In Table 4, the orthogonal (verimax) and oblique (oblimin) rotated factor loadings for the SBSoDDK and SBSoDGB are shown.

Table 5 shows the factor loadings for the SBSoDUS, SBSoDDE, SBSoDJP, SBSoDCN, and SBSoDTR, which, as shown in Table 2, all exhibited a one-factor structure.

As shown in Table 6, the correlations between the factor loadings for the SBSoDUS, SBSoDDE, SBSoDJP, SBSoDCN, and SBSoDTR were positive, significant, and moderate to strong. It was not possible to calculate Pearson’s r correlations between the factor loadings for all the SBSoD versions because the SBSoDDK and SBSoDGB exhibited a two-factor structure.

Table 7 shows the mean scores and standard deviations for the items of the SBSoDDK, SBSoDUS, SBSoDDE, SBSoDJP, SBSoDCN, SBSoDTR, and SBSoDGB.

In Table 8, the correlations between the mean scores for the items of the SBSoDDK, SBSoDUS, SBSoDDE, SBSoDJP, SBSoDCN, SBSoDTR, and SBSoDGB are shown. All of them, except for the ones between the mean scores for the items of the SBSoDCN and SBSoDGB and between the mean scores for the items of the SBSoDTR and the remaining SBSoD versions, were positive, significant, and moderate to strong.

Table 9 shows the mean total scores and standard deviations for the SBSoDDK, SBSoDUS, SBSoDDE, SBSoDJP, SBSoDCN, SBSoDTR, and SBSoDGB, both separated by gender (females and males) and combined. The mean total scores for the SBSoDDK, SBSoDUS, SBSoDJP, and SBSoDCN were significantly lower for females than for males.

## 4. Discussion

The primary and secondary aims of the current study were to validate the SBSoDDK with a focus on reliability, validity, and factor structure and compare performance on the SBSoD across samples with different nationalities, respectively. As we will discuss below, all the hypotheses were supported—at least partly—by the results.

### 4.1. The Validation of the SBSoDDK

#### 4.1.1. Reliability

Comparable to the SBSoDUS, SBSoDDE, SBSoDJP, SBSoDCN, SBSoDTR, and SBSoDGB, Cronbach’s alpha for the SBSoDDK reached acceptable levels, thereby affirming the internal consistency for the SBSoDDK. In addition, the correlations between the scores for each SBSoDDK item and the total SBSoDDK scores were positive, significant, and moderate to strong. This suggests that all the SBSoDDK items assess the same construct (the sense of direction). Therefore, the results supported hypotheses 1 and 2, and the SBSoDDK was found to be reliable. While Cronbach’s alphas for the 4MTper and 4MTmem only nearly reached acceptable levels, Cronbach’s alpha for the CFMT reached acceptable levels. Hence, hypothesis 3 was partly supported by the results. Although the lower Cronbach’s alphas for the 4MTper and 4MTmem do not affect the reliability of the SBSoDDK, they affected the estimation of the convergent validity of the SBSoDDK. However, this was partially addressed by calculating adjusted Pearson’s r correlations between the total SBSoDDK scores and the number of correct trials for the 4MTper and 4MTmem.

#### 4.1.2. Validity

The correlations between the total SBSoDDK scores and the number of correct trials for the 4MTper and 4MTmem were positive, significant, and weak to moderate, thereby affirming the convergent validity of the SBSoDDK. This aligns with other studies (e.g., [16]) demonstrating that the total SBSoDUS scores do not correlate strongly with performance on computer-based tests. The correlation between the total SBSoDDK scores and the number of correct trials for the CFMT was not significant, thereby affirming the discriminant validity of the SBSoDDK. Moreover, the correlations between the total SBSoDDK scores and the number of correct trials for the 4MTper and 4MTmem were significantly stronger than the correlation between the total SBSoDDK scores and the number of correct trials for the CFMT. Thus, the results supported hypotheses 4, 5, and 6, and the SBSoDDK was found to be valid.

#### 4.1.3. Factor Structure

Comparable to the SBSoDGB, the SBSoDDK exhibited a two-factor structure. This implies that for the Danish and British samples, the sense of direction is a construct with two dimensions. Conversely, the SBSoDUS, SBSoDDE, SBSoDJP, SBSoDCN, and SBSoDTR exhibited a one-factor structure. Furthermore, the correlations between the factor loadings for the SBSoDUS, SBSoDDE, SBSoDJP, SBSoDCN, and SBSoDTR were positive, significant, and moderate to strong. Together, this implies that for the American, German, Japanese, Chinese, and Turkish samples, the sense of direction is a construct with one, similar dimension. Therefore, parts 1 and 2 of hypothesis 7 were supported by the results. The differences in the factor structures for the different SBSoD versions may stem from the timing of their validation. The SBSoDDK and SBSoDGB were validated more recently than the SBSoDUS, SBSoDDE, SBSoDJP, SBSoDCN, and SBSoDTR, thereby making them more susceptible to the increasing use of GPSs. Indeed, studies (e.g., [26]) suggest that GPS usage can impair certain spatial abilities, such as mental rotation, which is crucial for survey strategies ([27]). This may contribute to a greater distinction between route and survey strategies and, consequently, the emergence of the two-factor structure.

It should be noted that for the SBSoDDK and SBSoDGB, factor 1 explained substantially more of the variance than factor 2. This suggests that even for the Danish and British samples, one of the dimensions dominates. In addition, for the SBSoDDK and SBSoDGB, factors 1 and 2 can be argued to reflect route and survey strategies. The SBSoDDK items that loaded strongly on factor 1 were either more representative of route strategies than of survey strategies (numbers 6 and 14) or representative of both route and survey strategies (numbers 1, 4, and 15). In contrast, the SBSoDDK items that loaded strongly on factor 2 were more representative of survey strategies than of route strategies (numbers 7 and 9). Similarly, the SBSoDGB items that loaded strongly on factor 1 were either more representative of survey strategies than of route strategies (numbers 7, 9, and 13) or representative of both route and survey strategies (number 1). In contrast, the SBSoDGB items that loaded strongly on factor 2 were more representative of route strategies than of survey strategies (numbers 10 and 14). Hence, for the Danish and British samples, the dominating dimensions seem to reflect route and survey strategies, respectively.

In studies ([18]; [32]) exploring whether and how mental maps are formed, mental maps are conceptualized as resembling conventional maps but with less detail and precision. With this conceptualization, the items of the SBSoDDK and SBSoDGB that address the capacity for forming mental maps (e.g., number 15) are more representative of survey strategies than of route strategies. However, this does not align with the fact that they loaded heaviest on the factor that can be argued to represent route strategies for both the SBSoDDK and SBSoDGB. As previously highlighted by [6] ([6]), mental maps can also be understood as knowledge of landmarks and their connections, which seems to be the case for the Danish and British samples. Thus, there is a discrepancy between how mental maps are conceptualized in the abovementioned studies and how they seem to be understood by the Danish and British samples.

[29] ([29]) recommends dividing any scale that assesses a construct with two or more dimensions into subscales. However, [9] ([9]) points out that this recommendation should be followed only if the dimensions are not equally important to the construct. As both route and survey strategies can be utilized when orienting oneself in the environment ([25]), they appear to be equally important to the sense of direction. Moreover, to allow for the comparison of performance on the SBSoD across the samples, the SBSoDDK and SBSoDGB were not divided into subscales.

### 4.2. A Comparison of Performance on the SBSoD Across Samples with Different Nationalities

With few exceptions, the correlations between the mean scores for the items of the different SBSoD versions were positive, significant, and moderate to strong. Therefore, part 3 of hypothesis 7 was partly supported by the results. Most of the exceptions were the correlations between the mean scores for the items of the SBSoDTR and the remaining SBSoD versions, which were all negative. Hence, there seems to be fewest similarities in how the Turkish and the remaining samples orient themselves in the environment. Furthermore, the mean total scores for the SBSoDUS, SBSoDDE, and SBSoDCN were higher than the ones for the SBSoDDK, SBSoDJP, SBSoDTR, and SBSoDGB. If taken at face value, this suggests that the American, German, and Chinese samples have a better sense of direction than the Danish, Japanese, Turkish, and British samples. However, even though individuals can generally judge their sense of direction well ([2]), how they do so varies across cultures. A study by [31] ([31]) showed that German and Turkish individuals tend to overestimate their sense of direction, whereas Danish individuals tend to underestimate their sense of direction. Thus, the differences in the mean total scores for the different SBSoD versions may be linked to variations in how individuals generally judge their sense of direction across cultures.

On the assumption that the samples do not have an equally good sense of direction, the differences in the mean total scores for the different SBSoD versions may also be linked to variations in the utilization of route and survey strategies across cultures. Specifically, survey strategies are generally considered superior to route strategies ([25]), and the utilization of route and survey strategies appears to vary across cultures. For instance, words from the absolute system, which are associated with survey strategies ([21]), seem to be used frequently in the United States and China ([22]), and the mean total scores for the SBSoDUS and SBSoDCN were higher than the ones for the SBSoDDK, SBSoDJP, SBSoDTR, and SBSoDGB. In addition, words for cardinal directions are from the absolute system ([21]), and the mean scores for the items of the SBSoDUS and SBSoDCN that address the experience with using cardinal directions (number 5) were higher than the ones for the corresponding items of the remaining SBSoD versions. However, in general, the mean scores for the items of the SBSoDUS and SBSoDCN were higher than for the ones for the items of the remaining SBSoD versions, including all the SBSoD items that are more representative of survey strategies than of route strategies (numbers 5, 7, 9, and 13). Furthermore, mental maps, which survey strategies rely on ([25]), are proposed to be formed more easily in a structured environment, which the environment in China and the United States tends to be, than in an unstructured one, which the environment in Denmark, Japan, Turkey, and Great Britain tends to be ([10]; [1]; [4]).

The interpretation of the differences in the mean total scores for the different SBSoD versions is complicated by the fact that the gender compositions of the samples vary. The gender compositions of the Danish, American, Japanese, Turkish, and British samples are skewed in favor of females and the gender composition of the Chinese sample is skewed in favor of males. Despite this, the mean total scores for the majority of the different SBSoD versions (the SBSoDDK, SBSoDUS, SBSoDJP, and SBSoDCN) were significantly lower for females than for males. Thus, part 4 of hypothesis 7 was partly supported by the results. Following this, it can be speculated that the mean total scores for the SBSoDDK, SBSoDUS, SBSoDJP, SBSoDTR, and SBSoDGB and the mean total score for the SBSoDCN were higher and lower, respectively, than they would have been if the gender compositions of the samples had not been skewed. In addition, it should be taken into consideration that females tend to underestimate their sense of direction, whereas males tend to overestimate their sense of direction. For example, a study by [23] ([23]) found that females reported a poorer sense of direction than males but performed as well as males on different relevant tasks. As a result, gender differences in the sense of direction may be exaggerated when assessed solely through self-report scales. This highlights the importance of considering biases in self-report scales and, ideally, employing relevant tasks alongside self-report scales to provide a more accurate and nuanced understanding of the sense of direction.

### 4.3. Weaknesses and Future Studies

The current study has certain weaknesses. Even though the translation of the SBSoDUS into Danish was based on expert review, back-translation was not used. Furthermore, cognitive interviews were only conducted when the SBSoDUS was translated into Turkish ([30]). Hence, it is possible that the samples varied in their understanding of (some of) the SBSoD items (though one could argue that this is also related to culture). Moreover, using nationalities as a proxy for culture in the current study was not optimal, and potential environmental heterogeneity within Denmark, the United States, Germany, Japan, China, Turkey, and Great Britain may not have been captured adequately. For instance, in the United States and Great Britain, some regions of the environment are unstructured, while other regions of the environment are structured ([7]; [20]). Finally, variations in the gender compositions of the samples complicated the interpretation of the differences in the mean total scores for the different SBSoD versions. Although the Danish sample was not representative of the Danish population, as it consisted of Danish university students and, thus, reflects a WEIRD group of people ([17]), it can also be seen as advantageous. The samples that were used for the studies ([22]; [30]; [6]) validating other versions of the SBSoD also consisted of university students (with different nationalities). If the samples had varied in other aspects than nationality (and gender), comparing performance on the SBSoD across the samples would have become more complicated. In addition, granting that variability is generally considered beneficial for correlation-based analyses, it is positive that the internal consistency and convergent and discriminant validity of the SBSoDDK were affirmed even when the Danish sample was homogeneous.

Future studies should focus on addressing the current study’s weaknesses as well as revising and further validating the SBSoD. Specifically, not all the SBSoD items appear to be equally relevant. The SBSoD items that address the capacity for finding one’s things (number 2) and the capacity for and attitude toward reading maps (numbers 7 and 9) are obvious candidates for revision. For example, there is a widespread use of GPSs nowadays, and studies (e.g., [5]) indicate that GPS usage can impact the sense of direction.

## 5. Conclusions

The SBSoDDK, which exhibited a two-factor structure, was found to be reliable and valid. This establishes it as a promising tool for the assessment of the sense of direction among the Danish population. The current study identified differences in performance on the SBSoD across samples with different nationalities, which may be linked to cultural variations in the sense of direction. However, the results need to be replicated in more representative and heterogeneous samples with less skewed gender compositions.

## Figures and Tables

**Figure 1 behavsci-15-00334-f001:**
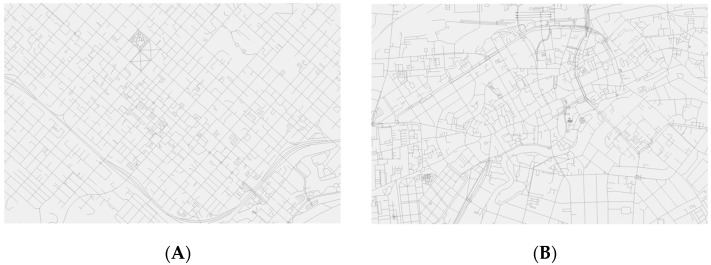
Generated using the OpenStreetMap, which is available under the Open Data Commons Open Database License by the OpenStreetMap Foundation. (**A**) An example of a structured environment in the United States; (**B**) an example of an unstructured environment in Denmark.

**Table 1 behavsci-15-00334-t001:** Sizes and gender compositions of the American, German, Japanese, Chinese, Turkish, and British samples.

Study	SBSoD Version	n	Gender Composition
[22] ([22])	SBSoDUS	106	(45% F, 53% M, 2% U)
	SBSoDDE	202	(69% F, 31% M)
	SBSoDJP	137	(63% F, 37% M)
	SBSoDCN	101	(28% F, 72% M)
[30] ([30])	SBSoDTR	251	(59% F, 41% M)
[6] ([6])	SBSoDGB	132	(94% F, 6% M)

Note. The SBSoDUS/DE/JP/CN/TR/GB = the American English/German/Japanese/Chinese/Turkish/British English versions of the Santa Barbara Sense of Direction scale. F = Females; M = Males; U = Unknown.

**Table 2 behavsci-15-00334-t002:** Internal consistency and factor structures for the SBSoDUS, SBSoDDE, SBSoDJP, SBSoDCN, SBSoDTR, and SBSoDGB.

Study	SBSoD Version	Internal Consistency ^1^	Factor Structure
[22] ([22])	SBSoDUS	0.88	1 ^2^
	SBSoDDE	0.86	1 ^2^
	SBSoDJP	0.89	1 ^2^
	SBSoDCN	0.80	1 ^2^
[30] ([30])	SBSoDTR	0.89	1 ^2,3^
[6] ([6])	SBSoDGB	0.87	2 ^2^

Note. The SBSoDUS/DE/JP/CN/TR/GB = the American English/German/Japanese/Chinese/Turkish/British English versions of the Santa Barbara Sense of Direction scale. ^1^ Measure: Cronbach’s alpha. ^2^ Explorative factor analysis conducted. ^3^ Confirmatory factor analysis conducted.

**Table 3 behavsci-15-00334-t003:** The item–total correlations for the SBSoDDK (n = 119).

Item	Pearson’s r	*p*, One-Tailed
1 (capacity for giving directions)	0.81	<0.001
2 (capacity for finding one’s things)	0.43	<0.001
3 (capacity for judging distances)	0.44	<0.001
4 (capacity for orienting oneself in the environment)	0.77	<0.001
5 (experience with using cardinal directions)	0.41	<0.001
6 (capacity for navigating in a new environment)	0.73	<0.001
7 (attitude toward reading maps)	0.67	<0.001
8 (capacity for understanding directions)	0.63	<0.001
9 (capacity for reading maps)	0.73	<0.001
10 (capacity for remembering routes)	0.47	<0.001
11 (attitude toward giving directions)	0.60	<0.001
12 (attitude toward knowing one’s position)	0.50	<0.001
13 (experience with navigational planning)	0.68	<0.001
14 (experience with remembering new routes)	0.76	<0.001
15 (capacity for forming mental maps)	0.79	<0.001

Note. The SBSoDDK = the Danish version of the Santa Barbara Sense of Direction scale.

**Table 4 behavsci-15-00334-t004:** The orthogonal (verimax) and oblique (oblimin) rotated factor loadings for the SBSoDDK and SBSoDGB.

Item	SBSoDDK				SBSoDGB			
	Orthogonal (Verimax)		Oblique (Oblimin)		Orthogonal (Verimax)		Oblique (Oblimin)	
	Factors				Factors			
	1	2	1	2	1	2	1	2
1 (capacity for giving directions)	0.72	0.41	0.72		0.63		0.66	
2 (capacity for finding one’s things)								
3 (capacity for judging distances)					0.50		0.52	
4 (capacity for orienting oneself in the environment)	0.72	0.34	0.74		0.59	0.57	0.57	0.39
5 (experience with using cardinal directions)		0.48		0.51	0.45		0.48	
6 (capacity for navigating in a new environment)	0.81		0.90		0.43	0.36	0.42	
7 (attitude toward reading maps)		0.83		0.84	0.76		0.84	
8 (capacity for understanding directions)	0.39	0.45	0.33	0.36	0.51	0.34	0.51	
9 (capacity for reading maps)		0.84		0.82	0.86		0.94	
10 (capacity for remembering routes)	0.47		0.53			0.72		0.76
11 (attitude toward giving directions)	0.48	0.30	0.47		0.48		0.50	
12 (attitude toward knowing one’s position)		0.40		0.35	0.32		0.32	
13 (experience with navigational planning)	0.52	0.38	0.50		0.59	0.35	0.60	
14 (experience with remembering new routes)	0.72		0.74			0.67		0.64
15 (capacity for forming mental maps)	0.81		0.87			0.56		0.52

Note: The SBSoDDK/GB = the Danish/British English versions of the Santa Barbara Sense of Direction scale. Empty fields: Factor loadings < 0.30.

**Table 5 behavsci-15-00334-t005:** The factor loadings for the SBSoDUS, SBSoDDE, SBSoDJP, SBSoDCN, and SBSoDTR.

Item	SBSoDUS	SBSoDDE	SBSoDJP	SBSoDCN	SBSoDTR
1 (capacity for giving directions)	0.77	0.62	0.79	0.79	0.73
2 (capacity for finding one’s things)	0.44	0.21	0.18	0.24	0.38
3 (capacity for judging distances)	0.48	0.33	0.75	0.48	0.60
4 (capacity for orienting oneself in the environment)	0.73	0.79	0.76	0.84	0.79
5 (experience with using cardinal directions)	0.55	0.37	0.54	0.48	0.59
6 (capacity for navigating in a new environment)	0.45	0.71	0.46	0.64	0.65
7 (attitude toward reading maps)	0.64	0.62	0.75	0.37	0.54
8 (capacity for understanding directions)	0.54	0.54	0.71	0.41	0.59
9 (capacity for reading maps)	0.75	0.67	0.82	0.59	0.63
10 (capacity for remembering routes)	0.55	0.43	0.54	0.53	0.65
11 (attitude toward giving directions)	0.47	0.62	0.62	0.40	0.49
12 (attitude toward knowing one’s position)	0.51	0.32	0.33	0.17	0.30
13 (experience with navigational planning)	0.69	0.72	0.44	0.47	0.52
14 (experience with remembering new routes)	0.59	0.63	0.68	0.22	0.75
15 (capacity for forming mental maps)	0.56	0.58	0.63	0.27	0.64

Note. The SBSoDUS/DE/JP/CN/TR = the American English/German/Japanese/Chinese/Turkish versions of the Santa Barbara Sense of Direction scale.

**Table 6 behavsci-15-00334-t006:** The correlations between the factor loadings for the SBSoDUS, SBSoDDE, SBSoDJP, SBSoDCN, and SBSoDTR.

	SBSoDUS	SBSoDDE	SBSoDJP	SBSoDCN
SBSoDDE	0.60 **			
SBSoDJP	0.59 *	0.53 *		
SBSoDCN	0.56 *	0.53 *	0.49 *	
SBSoDTR	0.51 *	0.61 **	0.71 **	0.64 **

Note. The SBSoDUS/DE/JP/CN/TR = The American English/German/Japanese/Chinese/Turkish versions of the Santa Barbara Sense of Direction scale. Significance level: * *p* < 0.01, ** *p* < 0.05.

**Table 7 behavsci-15-00334-t007:** The mean scores and standard deviations for the items of the SBSoDDK, SBSoDUS, SBSoDDE, SBSoDJP, SBSoDCN, SBSoDTR, and SBSoDGB.

Item	SBSoDDK	SBSoDUS	SBSoDDE	SBSoDJP	SBSoDCN	SBSoDTR	SBSoDGB
1	3.6 (1.7)	4.8 (1.6)	4.4 (1.5)	3.5 (1.5)	4.4 (1.9)	3.1 (1.6)	3.9 (1.7)
2	3.7 (1.7)	4.3 (1.9)	4.6 (1.8)	3.3 (1.7)	3.8 (2.1)	3.2 (1.9)	4.3 (1.7)
3	3.8 (1.7)	4.4 (1.6)	3.6 (1.7)	3.4 (1.7)	4.1 (1.7)	3.3 (1.6)	3.7 (1.6)
4	3.9 (1.8)	4.8 (1.9)	4.3 (1.8)	3.3 (1.7)	4.0 (2.1)	3.2 (1.7)	4.0 (1.8)
5	2.4 (1.9)	3.6 (2.3)	2.3 (1.6)	2.4 (1.8)	3.9 (2.6)	3.3 (1.7)	2.2 (1.5)
6	3.6 (1.9)	4.0 (1.6)	4.3 (1.8)	3.9 (1.9)	4.0 (2.1)	3.4 (1.8)	3.7 (1.8)
7	3.4 (1.8)	4.2 (2.0)	4.2 (2.0)	4.0 (2.1)	5.6 (2.0)	3.7 (1.9)	3.1 (1.8)
8	4.2 (1.8)	5.1 (1.6)	5.0 (1.4)	3.6 (1.8)	4.8 (2.0)	2.9 (1.7)	4.1 (1.7)
9	3.9 (1.6)	5.0 (1.6)	4.6 (1.7)	3.7 (1.8)	5.2 (1.8)	3.8 (1.6)	3.4 (1.8)
10	3.3 (1.9)	4.1 (2.1)	3.5 (2.0)	3.5 (1.9)	3.5 (2.0)	3.2 (1.8)	4.0 (2.0)
11	3.7 (1.8)	4.0 (1.7)	4.1 (1.9)	3.6 (1.6)	4.5 (1.8)	3.3 (1.9)	3.4 (1.6)
12	4.7 (1.7)	6.0 (1.4)	5.3 (1.6)	5.1 (1.5)	5.7 (1.8)	2.0 (1.6)	5.2 (1.6)
13	3.9 (2.0)	4.4 (2.1)	4.4 (2.1)	4.0 (2.0)	4.0 (2.1)	3.1 (1.9)	3.8 (2.0)
14	3.7 (1.9)	5.0 (1.7)	4.4 (1.9)	3.3 (1.6)	4.4 (1.9)	3.0 (1.7)	4.1 (1.8)
15	4.3 (1.7)	5.3 (1.8)	4.6 (1.8)	3.7 (1.6)	4.5 (2.0)	3.1 (1.8)	4.2 (1.8)

Note. The SBSoDDK/US/DE/JP/CN/TR/GB = the Danish/American English/German/Japanese/Chinese/Turkish/British English versions of the Santa Barbara Sense of Direction scale.

**Table 8 behavsci-15-00334-t008:** The correlations between the mean scores for the items of the SBSoDDK, SBSoDUS, SBSoDDE, SBSoDJP, SBSoDCN, SBSoDTR, and SBSoDGB.

	SBSoDDK	SBSoDUS	SBSoDDE	SBSoDJP	SBSoDCN	SBSoDTR
SBSoDUS	0.85 **					
SBSoDDE	0.89 **	0.77 **				
SBSoDJP	0.75 **	0.61 **	0.74 **			
SBSoDCN	0.47 *	0.58 *	0.54 *	0.64 **		
SBSoDTR	−0.55 *	−0.64 **	−0.42	−0.50 *	−0.19	
SBSoDGB	0.84 **	0.76 **	0.78 **	0.66 **	0.18	−0.73 **

Note. The SBSoDDK/US/DE/JP/CN/TR/GB = the Danish/American English/German/Japanese/Chinese/Turkish/British English versions of the Santa Barbara Sense of Direction scale. Significance level: * *p* < 0.01, ** *p* < 0.05.

**Table 9 behavsci-15-00334-t009:** The mean total scores and standard deviations for the SBSoDDK, SBSoDUS, SBSoDDE, SBSoDJP, SBSoDCN, SBSoDTR, and SBSoDGB, both separated by gender (females and males) and combined.

	Total Score	Total Score for F	Total Score for M	Cohen’s d
SBSoDDK	3.7 (1.1)	3.5 (1.1) ***	4.3 (1.0)	0.8
SBSoDUS	4.6 (1.1)	4.3 (1.2) **	4.9 (0.9)	0.5
SBSoDDE	4.2 (1.0)	4.2 (1.0)	4.3 (1.0)	
SBSoDJP	3.6 (1.1)	3.3 (1.0) ***	4.2 (1.0)	0.9
SBSoDCN	4.4 (1.0)	4.1 (1.1) *	4.6 (1.0)	0.5
SBSoDTR ^1^	3.2 (1.1)			
SBSoDGB ^1^	3.8 (1.0)			

Note. The SBSoDDK/US/DE/JP/CN/TR/GB = the Danish/American English/German/Japanese/Chinese/Turkish/British English versions of the Santa Barbara Sense of Direction scale. F = Females; M = Males. Significance level: *** *p* < 0.001, ** *p* < 0.01, * *p* < 0.05, one-tailed. ^1^ It was not possible to separate the mean total scores by gender.

## Data Availability

Data for the SBSoDDK, 4MT, and CFMT are available in a file ([13]) on the Open Science Framework.

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
