# Peer review of "The Validation of the Danish Version of the Santa Barbara Sense of Direction Scale (SBSoDDK) and a Comparison of Performance on the SBSoD Across Samples with Different Nationalities"

_behavsci, 2025, doi:10.3390/bs15030334_

Round 1
Reviewer 1 Report
Comments and Suggestions for Authors
Manuscript Number: behavsci-3456236
Title: Validation of the Danish Version of the Santa Barbara Sense of Direction (SBSoDDK) and Comparison of Performance on the SBSoD across Samples With Different Nationalities
Major revision
Some important steps are missing in the validation process of a tool:
- Back-translation
- Expert review: A group of experts in psychology and methodology discusses the differences and adapts the terms to ensure semantic and cultural equivalence.
- Test on a small sample (N=20-50): The translated version is administered to a small group of individuals with characteristics similar to the target population to identify comprehension difficulties or ambiguities.
- Cognitive interviews: Participants are asked to explain their reasoning behind their responses to verify the correct interpretation of the items. Unlike the Turkish version, where cognitive interviews were conducted to assess item comprehension, the Danish version lacks this step. This raises concerns about whether participants interpreted items consistently
Moreover:
The study indicates that gender distributions vary significantly across samples, complicating cross-national comparisons. The interpretation of differences in navigation ability may be influenced more by gender than by cultural factors. Indeed, the study relies on self-reported sense of direction, which can be prone to biases such as overestimation or underestimation of one’s abilities. Specifically, women underestimate their sense of direction (e.g., Nori and Piccardi, 2015)
The description of the participants does not include age, gender (male and female), or dominant hand, which can be essential factors to consider in the field of spatial cognition.
The Danish version of the Santa Barbara Sense of Direction (SBSoDDK) revealed a two-factor structure, while other versions (e.g., American, German, Chinese) retained a one-factor structure. I suggest the authors to explain better this difference
The study does not address how the increasing use of GPS technology may have altered individuals’ reliance on mental maps and spatial strategies. This could affect the validity of comparing older and newer datasets. I suggest the authors analyse some works about the relationship between spatial navigation and GPS, such as Ishikawa et al., Ruginski et al., Dahmani and Bohbot, He and Hegarty, Nori et al.
Some statistical analyses (e.g., Cronbach’s alpha for certain tests) indicate only marginally acceptable reliability. While the scale is deemed valid, refinements may be necessary for stronger psychometric robustness.
I suggest the authors to insert ethical approval by them University
Reviewer 2 Report
Comments and Suggestions for Authors
In general, I have read this manuscript with interest and I think the quality of the writing (both with regard to the English language and the readability) is high to very high. I found the way in which the hypotheses have been concretely stated in 2.3 as well as the stepwise discussion of the hypotheses in the Results and Discussion sections very clear. All Tables provide relevant information in a well structured manner. With regard to the first part/aim of the manuscript, I think that the authors have provided convincing evidence that the Danish version of the SBSoD is valid, reliable and shows significant overlap with other language versions of this scale. My comments and concerns rather relate to the second part/aim of the manuscript, i.e. the comparison of the SBSoD-scores across samples with multiple nationalities. Due to the rather speculative nature of this part of the manuscript, I have some doubts about its added value to the manuscript.
- General, major comment: Although the authors do acknowledge that there are major differences in male/female ratios across the samples from the different nationalities, I think that these differences are so pronounced and crucial (e.g., the Chinese and British samples are extremely different from the others) and wonder whether making male-female comparison is actually justified based on these very different samples.
- Major comment: The authors state, with a reference to a study by Clack and Maguire from 2020, that self-reported sense of direction can more or less be interpreted as reflective of actual sense of direction. In (neuro)psychology, however, there is a wide range of studies showing that people have difficulty assessing their actual cognitive abilities. A recent example of such a study with regard to navigation ability in particular is a study by Van der Ham, Van der Kuil and Claessen from 2021 in Aging & Mental Health. They indicated in a very large sample that people are able to provide some indicative self-reported evaluation of their actual navigation abilities, however, these self-reports were also subject to strong systematic biases related to age and gender (more specifically an interaction between these two factors). I think that authors need to acknowledge more explicitly that self-reported sense of direction might provide an indication of this ability but is not the same as a direct, objective assessment of sense of direction.
- Major comment: The authors report that the mean total scores for the US, German and Chinese versions are higher than the other language versions of the SBSoD. They interpret this as an indication for survey knowledge being better developed in US and Chinese inhabitants those from the other countries. Is this also reflected in higher means of the specific SBSoD items that reflect survey knowledge in particular as interpreted by the authors (i.e., items 5, 7, 9, 13) or are simply all item scores higher in these US and Chinese samples? (I.e., does it have to do with survey knowledge in particular or are scores on all items simply higher in these countries?) And what explanation do the authors have for the higher score in the German sample? As far as I know, German layouts are not as structured as is the case in the US or China.
- Minor comment: The authors write in line 378 that the scores on the SBSoD indicate that "females have a poorer sense of direction than males". In my view, it indicates that females subjectively report having a poorer sense of direction (which is not the same). E.g., the study by Van der Ham and colleagues (2021) indicated that especially younger females (i.e., including the typical age range of university students) tend to underestimate their actual navigation abilities. Males of the same age range do exactly the opposite (i.e., they significantly overestimate their actual abilities).
- Minor comment: I have some difficulties understanding the statement in lines 379-382 "[...] were higher and lower, respectively, than they would have been if the gender compositions for the samples had not been skewed". Is this statement based on speculation or on some statistical correction/procedure?
Round 2
Reviewer 1 Report
Comments and Suggestions for Authors
Manuscript Number: behavsci-3456236
Title: Validation of the Danish Version of the Santa Barbara Sense of Direction (SBSoDDK) and Comparison of Performance on the SBSoD across Samples With Different Nationalities
A few doubts remain regarding the study.
First, there is concern that age may negatively impact visuospatial abilities, which could, in turn, affect Sense of Direction (SoD). It would have been beneficial to limit the participant age range to 20-25 years and exclude the 10% of participants with a median age of 50. Additionally,
I recommend that the authors elaborate on the issue of overestimating and underestimating one's Sense of Direction. (e.g., I believe I'm good at orienting myself... But is that true? Nori, R., & Piccardi, L. Cognitive Processing, 2015, 16(3), pp. 301-307)
Reviewer 2 Report
Comments and Suggestions for Authors
I would like to thank the authors for carefully considering my comments and suggestions. I think the fast majority of them has been handled well in the author's response and by way of additions/reformulations in the revised version of the manuscript.
My only point related to this revised version of the manuscript is connected to my previous question about the higher scores in the US and Chinese sample related to survey knowledge. Based on the data in Table 7, it seems that the US and Chinese samples score (much) higher on nearly all items. Are the authors sure that the difference for the items that are typical of survey knowledge (5, 7, 9, 13) are indeed significantly higher than for the items that are typical of route knowledge? As this is now what they now state in the manuscript in section 4.2.
